# Recent Possibilities for the Diagnosis of Early Pregnancy and Embryonic Mortality in Dairy Cows

**DOI:** 10.3390/ani11061666

**Published:** 2021-06-03

**Authors:** Ottó Szenci

**Affiliations:** Department of Obstetrics and Food Animal Medicine Clinic, University of Veterinary Medicine Budapest, H-2225 Üllő-Dóra Major, Hungary; szenci.otto@univet.hu

**Keywords:** dairy cow, ultrasonography, pregnancy proteins, embryonic mortality, fetal mortality

## Abstract

**Simple Summary:**

Pregnancy diagnosis plays an essential role in decreasing days open in dairy farms; therefore, it is very important to select an accurate method for diagnosing early pregnancy. Besides traditional pregnancy diagnoses made by rectal palpation of the uterus from 40 to 60 days after AI and measuring the serum or milk progesterone concentration between 18 to 24 days after AI, there are several new possibilities to diagnose early pregnancy in dairy farms. However, it is very important to emphasize that before introducing any new diagnostic test, we need to make sure the accuracy of that particular test to be able to decrease the rate of iatrogenic pregnancy losses caused by prostaglandin or resynchronization treatments. This review focuses on the diagnostic possibilities and limitations of early pregnancy diagnosis in the field.

**Abstract:**

One of the most recent techniques for the on-farm diagnosis of early pregnancy (EP) in cattle is B-mode ultrasonography. Under field conditions, acceptable results may be achieved with ultrasonography from Days 25 to 30 post-AI. The reliability of the test greatly depends on the frequency of the transducer used, the skill of the examiner, the criterion used for a positive pregnancy diagnosis (PD), and the position of the uterus in the pelvic inlet. Non-pregnant animals can be selected accurately by evaluating blood flow in the corpus luteum around Day 20 after AI, meaning we can substantially improve the reproductive efficiency of our herd. Pregnancy protein assays (PSPB, PAG-1, and PSP60 RIA, commercial ELISA or rapid visual ELISA tests) may provide an alternative method to ultrasonography for determining early pregnancy or late embryonic/early fetal mortality (LEM/EFM) in dairy cows. Although the early pregnancy factor is the earliest specific indicator of fertilization, at present, its detection is entirely dependent on the use of the rosette inhibition test; therefore, its use in the field needs further developments. Recently found biomarkers like interferon-tau stimulated genes or microRNAs may help us diagnose early pregnancy in dairy cows; however, these tests need further developments before their general use in the farms becomes possible.

## 1. Introduction

The extent of pregnancy loss (PL) in dairy cows can be estimated from the difference between the fertilization rate (FR) and the subsequent calving rate (CR). It is generally accepted that the FR in healthy dairy cattle is between 85% and 90% [1] or, more recently, around 80% [2]. The CR has decreased from 66% since 1951, to about 50% until 1975 [3], and furthermore to about 33.1% in Spain [4], 33.4% in Israel [5], 37% in Canada [6], 41% in Japan [7], and 40% to 44% in the USA [3,8], while it did not change (>60%) in heifers [9,10]. Therefore, 41% to 57% of pregnancies can be lost during gestation. Most of the losses (early embryonic mortality: EEM) may occur before the maternal recognition of gestation (<Day 16) when the life of the corpus luteum is not extended, and cows return to estrus [1,11]. After the maternal recognition of pregnancy, the life of the corpus luteum will be extended, and embryonic mortality (further 5% to 10%) occurring between Days 16 and 42 after AI is called late embryonic mortality (LEM), while occurring between Days 42 to 90 after AI is named early fetal mortality (EFM). Late fetal mortality between Day 90 and term is rare [12]. It has recently been reported that PL after the first month of gestation may range between 3.2% and 42.7% [13].

Recently, Pohler et al. [14] have suggested using the terms ‘early embryonic development’ until Day 30 and ‘late embryonic development’ between Days 31 and 45 of gestation. Due to the fact that different technical terms are used for this period, perhaps it would be easier to divide the former late embryonic period between Days 16 to 42 into two stages, namely, stage 1 between Days 16 and 30 and stage 2 between Days 31 and 42/45 of gestation.

The PL represents a considerable biological and economic waste for the farmer, therefore, the objective of the present review is to discuss recent methods suitable for diagnosing early pregnancy and pregnancy losses in dairy cattle.

## 2. Early Pregnancy Diagnosis (PD)

### 2.1. Real-Time B-Mode Ultrasonography

During ultrasonographic examination (UE), a cow was initially considered to be pregnant when an irregularly shaped, non-echogenic black spot (or spots) were recognized within the uterine lumen, representing the chorioallantoic fluid [15]. The demonstration of an embryo (later with a beating heart) and/or embryonic membranes provided additional confirmation of pregnancy [16,17]. A normal amount of chorioallantoic fluid in the uterine horn ipsilateral to the ovary with a corpus luteum provided additional confirmation of a normal gestation [17,18]. Where no such signs (Table 1) were found, the possibility of pregnancy was ruled out, giving a non-pregnancy diagnosis. The confirmation of ultrasonographic diagnoses was usually based on palpation per rectum of the uterus at 2 to 3 months post-AI, or upon spontaneous return to estrus after AI. A cow was also considered pregnant if an embryo proper with a beating heart was recognized at a final UE on Days 50 to 60 post-AI. Cows diagnosed as non-pregnant by palpation per rectum or by ultrasonography between Days 50 and 90 were usually designated as non-pregnant [19,20,21].

Under experimental conditions, embryonic vesicles in dairy cows can be detected as early as at 9 [23], 10 [20], or 12 days [24] of gestation; however, an accurate pregnancy diagnosis (100%) can be reached only on Days 20 and 22 of gestation [25], when the embryo (20.3 days) and its heartbeat (20.9 days) can be detected [21]. It is important to mention that fluid accumulation in the chorioallantois during early pregnancy can be easily confused with the uterine fluid within the uterus during pro-estrus and estrus [25].

Under on-farm conditions, between Days 22 to 49 after AI, the sensitivity, specificity, positive predictive value, and negative predictive value of ultrasonographic diagnoses made with a 5 MHz linear-array or sector transducer varied between 95% and 100%, 67% and 100%, 85.7% and 100%, and 84% and 100%, respectively (Table 2).

In contrast, Badtram et al. [30] reported that the sensitivity and specificity of the ultrasound test between Days 23 and 31 post-AI were only 68.8% and 71.7%, respectively. In a recent study, maximum sensitivity and negative predictive value were reached at Day 26 in dairy heifers and at Day 29 in dairy cows, while the specificity and positive predictive values were 96.7% and 94.4% for dairy heifers and 96.3% and 91.3% for dairy cows, respectively [17]. In contrast, according to Nation et al. [16], due to pregnancy loss, Days 28 to 35 after AI are too early for reliable detection of pregnancy in dairy cows.

Between Days 27 to 34 after AI, the sensitivity, specificity, positive predictive value, and negative predictive value of ultrasonographic diagnoses made by the use of 5–10 MHz transducer varied between 90% and 96.6%, 91.5% and 100%, 88.4% and 100%, and 92.3% and 97.8%, respectively (Table 3). When the recognition of an embryo proper with a beating heart was used as the criterion for a positive ultrasound PD, significantly (*p* < 0.001) more false negative and less false positive ultrasound diagnoses were made, in comparison with the recognition of chorioallantoic fluid [31]. In contrast, Silva et al. [18] reported that the presence of chorioallantoic fluid in the uterine lumen and a corpus luteum alone might lead to more false positive results than when an embryo was visualized by using a 5–10 MHz linear-array transducer on Day 27 after timed AI.

The reliability of the test greatly depends on the frequency of the transducer used, the skill of the operator [30,34], the criterion used for a positive pregnancy diagnosis (uterine fluid or embryo [15,27,32]; uterine fluid, embryonic membranes, or embryo [16]; amniotic fluid and/or embryo surrounded by an amniotic sac [29]; normal amount of chorioallantoic fluid, embryo, ipsilateral CL [18]; embryo with a heartbeat and corpus luteum [35]; embryo with a heartbeat, fluid-filled uterine horn, ipsilateral corpus luteum [36,37]; amniotic fluid, embryo, or embryonic heartbeat [38]; accumulation of intrauterine gestural fluid in non-pregnant cows [25], and the position of the uterus in the pelvic inlet [39]. More incorrect non-pregnancy diagnoses were made between Days 24 to 38 in cows in which the uterus was located far cranial to the pelvic inlet, in comparison with cows in which the uterus was located within or close to the pelvic inlet [39]. It is important to mention if our ultrasonographic pregnancy diagnoses are based on detection of uterine fluid in the uterus on Day 29 after timed AI, at that time, these cows can be classified 3.8 times more likely as not pregnant 74 days after timed AI than those cows diagnosed pregnant based on visualization of an embryo with a heartbeat [36].

Special attention must be paid to the diagnosis of twin pregnancy by clearly locating the two embryos because twin pregnancy loss and spontaneous twin reduction have been reported to occur up to Day 90 of gestation [40,41]. Possibilities for diagnosing twin pregnancy in the field have been reviewed recently by Szelényi et al. [42].

### 2.2. Color Flow Doppler Ultrasonography

Color flow Doppler ultrasonography (CFDU) can be used to monitor blood flow in the corpus luteum (CL) around Day 20 after AI because luteal vascularization plays a decisive role in the functional evaluation of the corpus luteum [43,44,45,46]. On the other hand, there is a good correlation between decreasing blood flow during CL regression and progesterone concentration [47,48]. One of the main advantages of CFDU is its high sensitivity (99%) and negative predictive value (98.5%) of diagnosing non-pregnant dairy cattle on Day 20 after AI, which results in few false negative diagnoses [49]. According to Dubuc et al. [50], by monitoring the blood flow in the CL on Day 21 after AI in contrast with measuring the progesterone concentration, non-pregnancy in dairy cows on Day 32 after AI can also be predicted with high accuracy (sensitivity: 99.8%, negative predictive value: 99.3%). In a recent study, Siqueira et al. [51] have found that the reduction of blood flow in the CL takes place some days before any detectable changes in CL morphology and echotexture, and therefore, determination of the blood flow and the adjusted blood flow calculated from the ratio of luteal tissue area and blood flow were the best early predictors of non-pregnancy. In contrast, luteal tissue area and echotexture were found to be inconsistent early indicators of luteolysis.

### 2.3. Conceptus Proteins

Trophoblastic mono- and binucleate cells from the early bovine conceptus synthesize substantial amounts of proteins. Among these, one has been described as bovine pregnancy-specific protein B (bPSPB), which enters into the maternal circulation [52]. In addition, a bPSPB-related protein, designated bovine pregnancy-associated glycoprotein (bPAG; [53]) or bPAG-1 [54], as well as pregnancy serum protein 60 kDa (PSP60; [55]) have been described. Pregnancy proteins are inactive members of the aspartic proteinase family (pepsins, cathepsins, and renins also belong to this family), and are identical in genetic nucleotide sequence [56,57,58]. The isolated preparations of pregnancy proteins may differ in carbohydrate and sialic acid content, which may explain their minor differences in profile and disappearance from the maternal circulation after calving or EM [32,59,60]. Because the concentrations of these proteins gradually increase during pregnancy, they are good indicators of the presence of a live embryo [61]. The pregnancy proteins have been detected in the serum of some pregnant cows as early as Days 15 to 22 [61,62] or Day 22 [55,63] after AI. Due to the delayed appearance of these proteins in the blood in some cows, their use for PD provides more accurate results from Days 28 to 30 onwards [32,57,64]. Both bPSPB and bPAG-1 have been detected in the peripheral circulation during the postpartum period 70 to 100 days after calving [61,65]. Likewise, Mialon et al. [55] reported similar residual protein concentrations for PSP60 after calving in the Charolais (mean: 107 days/91–119/), Normande (mean: 84 days/56–105/), and Holstein (mean: 88 days/63–126/) breeds. In a recent study, 56.7% and 44.9% of the false positive diagnoses based on bPSPB and bPAG-1 tests, respectively, originated from cows that were inseminated within 70 days after calving [32]. These findings indicate that the presence of bPSPB and bPAG-1 in the plasma of cows during the early stages of the postpartum period may limit their use under field conditions. If only those cows are selected for the protein tests, which are inseminated after Day 50 [66] or Day 70 after calving [67,68,69], post-calving interference with the residual bPSPB or bPAG-1 in the peripheral circulation during the postpartum period can be minimal.

A further limitation after late embryonic mortality (LEM) is that protein levels may remain above the threshold level, although the concentration of both proteins decreases steadily [70,71]. This is probably related to the relatively long half-life (7–8 days for bPSPB and 3–4 days for bPAG-1) in the maternal circulation after EM [70,72].

Serrano et al. [73] reported that the herd, fetal sex, milk production, lactation number, and plasma progesterone concentrations did not significantly influence the plasma PAG-1 concentration, while twin pregnancy, the use of Limousin semen and conception during the cool period significantly increased its concentrations throughout gestation. Clone pregnancies comparing with control cows may also significantly increase pregnancy protein concentrations during the whole gestation, regardless of pregnancy outcome [74].

#### 2.3.1. In-House PAGs ELISA Tests

Originally PSPB, PAG-1, and PSP60 radioimmunoassay tests were used to detect pregnancy proteins; however, in the meantime, it turned out that there are 21 PAG family members, and PAG-1 used for pregnancy diagnosis was not the earliest pregnancy protein produced by the trophoblast [75]. After recognizing this, the first in-house sandwich ELISA protocol was developed for measuring the circulating concentration of PAGs by Green et al. [76] and Friedrich and Holtz [77]. Between Days 26 and 58 after AI, the sensitivity, specificity, positive predictive value, and negative predictive value of the in-house PAGs ELISA tests varied between 94% and 100%, 77% and 94.2%, 90.7% and 97.8%, and 91.2% and 97.1%, respectively (Table 4).

Mercadante et al. [81] found higher PAGs concentrations (in-house ELISA) in primiparous cows compared with multiparous pregnant cows (in agreement with Ricci et al. [36], while Kaya et al. [82] found similar differences between heifers and lactating cows), during the second and later breeding compared with the first breeding postpartum, in cows experiencing clinical metritis, metabolic problems and left displaced abomasum after calving compared with cows not experiencing those clinical diseases and in cases of greater milk yield, while the body condition score did not influence it. According to Dufour et al. [38], the accuracy of the PAG test (commercial milk ELISA) was not influenced by parity, the number of days, since the last breeding, and the level of milk production, while Ricci et al. [37] found a negative correlation between plasma and milk PAG concentration (commercial ELISA) and milk production.

#### 2.3.2. Commercial ELISA Tests

There are currently several ELISA tests are available on the market, as discussed below.

BioPRYN ELISA test for the detection of pregnancy-specific protein B in the serum as early as 28 days after breeding in cattle with no interference from a previous pregnancy as early as 73 days after calving.

Between Days 26 to 58 after AI, the sensitivity, specificity, positive predictive value, and negative predictive value of the BioPRYN ELISA test varied between 93.9% and 100%, 87% and 97.1%, 92% and 99.3%, and 91.7% and 97.8%, respectively (Table 5).

Martins et al. [83] compared the basal serum PSPB concentrations determined at Day 17 after AI with Day 23 results. They found that if the difference was more than 28% at that time the sensitivity, and the specificity of the ELISA test were higher (98% and 97%), compared to 92.8% and 97% obtained when only Day 23 values were evaluated. It is also important to mention that the concentrations of PSPB on Days 23 and 28 were related to pregnancy losses between Days 28 and 35 after AI, but not to those occurring in later periods (Days 35 to 56 and Days >56 to calving).

Middleton and Pursley [84] suggested comparing the results of serum PSPB samples withdrawn on Days 17 and 24 after AI to diagnose non-pregnant cows with 100% accuracy at Day 24 after AI and to predict early pregnancy loss.

DG29^®^ Bovine Blood Pregnancy ELISA Test for the detection of specific pregnancy-related protein in serum as early as 29 days after breeding in cattle with no interference from a previous pregnancy as early as 90 days after calving. Between Days 28 and 40, after AI the sensitivity, specificity, positive predictive value, and negative predictive value of the D29 test varied between 90.2% and 100%, 66.7% and 98.3%, 91% and 97.4%, and 93.7% and 100%, respectively (Table 6).

The IDEXX Bovine Pregnancy ELISA Test for the detection of early pregnancy-associated glycoproteins in the serum or plasma of cattle as early as 28 days after breeding in cows with no interference from a previous pregnancy as early as 60 days after calving. Between Days 25 and ~41, after AI the sensitivity, specificity, positive predictive value, and negative predictive value of the test varied between 92% and 100%, 87% and 100%, 84% and 100%, and 94.2% and 100%, respectively (Table 7).

The IDEXX Milk Pregnancy ELISA Test for the detection of pregnancy-associated glycoproteins in bovine milk from 28 days after breeding in cows with no interference from a previous pregnancy as early as 60 days after calving. Between Days 28 to ~53, after AI the sensitivity, specificity, positive predictive value, and negative predictive value of the test varied between 96% and 100%, 83% and 97.9%, 79% and 98.5%, and 96% and 100%, respectively (Table 8). While on Day ≥60, the sensitivity and the specificity of the test varied between 98.5–99.2% and 95.5–96.7%, respectively, the positive predictive value and the negative predictive value were 99.8% and 80.8%, respectively (Table 8). It is important to mention that the plasma PAG levels turned out to be approximately twice higher than the milk PAG levels [36]. 

The Rapid Visual Pregnancy ELISA test has been recently developed to detect early pregnancy-associated glycoproteins in bovine whole blood, plasma, or serum as early as 28 days after breeding with no interference from a previous pregnancy as early as 60 days after calving.

This test can be run without ELISA instrumentation and read visually [97]. Between Days 25 and 45 after AI, the sensitivity, specificity, positive predictive value, and negative predictive value of the visual blood test for dairy cattle were 99.8%, 91.7%, 92.7%, and 99.7%, respectively [97]. When the IDEXX visual ELISA test was used, the sensitivity, specificity, positive predictive value, and negative predictive value of the whole blood test were 98 ± 1%, 85 ± 3%, 87 ± 3%, and 98 ± 1%, respectively [92]. The high sensitivity and negative predictive values mean that very few truly pregnant cows were misdiagnosed as not pregnant. The overall accuracy of the test was 92 ± 2% [92]. When a microtiter plate reader was used to measure the optical density for individual wells in the ELISA plate, the overall accuracy of the test became 94 ± 1% [92]. According to the agreement analysis, a very good agreement between visual and milk ELISA tests (kappa: 0.92) and between visual and serum ELISA tests (kappa: 0.97), respectively, were reported [34].

The high overall accuracy of the test (98.9%) was reported when the BioPRYN Rapid Visual Pregnancy Test^®^ was used in 92 dairy cattle on Day 28 in heifers and on Day 30 in cows. However, it is important to mention that there were three samples evaluated as ‘to be rechecked’; however, later, it was not possible to evaluate them, and therefore, these three samples were removed from the dataset [98]. A somewhat lower overall accuracy (90%) was reported when the Ubio quickVET rapid visual test was used for plasma samples between Days 30 and 40 after AI [90], and a much lower accuracy (70%) was obtained when the Fassisi^®^ BoviPreg visual test kit was used for serum samples on Day 30 after AI [99].

### 2.4. Early Pregnancy Factor (EPF)

The earliest specific indicator of fertilization and the continuing presence of a viable conceptus is a serum constituent that had originally been detected in mice [100]. This substance is known as the early pregnancy factor (EPF) and has also been described in women [101], sheep [102], cattle [103], and pigs [104].

The reported and extraordinary properties of EPF include:

Early appearance (within hours) after mating or insemination

Rapid disappearance following induced death or removal of the embryos [105,106].

These factors suggest that EPF may be the most useful tool for investigating early embryonic survival or failure [106,107,108]. According to Laleh et al. [109], the rosette inhibition test (RIT) has the potential to distinguish pregnant from non-pregnant dairy cows in the first week of pregnancy. However, the detection of EPF is entirely dependent on a bioassay that is not practical. After identifying immunosuppressive EPF with a molecular weight of approximately 200,000 and raising polyclonal antibodies against it [110], a new diagnostic test, the early conceptus factor (ECF) test, was developed in the USA for field use; however, it cannot accurately identify conception within days or at any time before Day 21 of gestation [111,112,113,114,115].

After EPF, a 10.84 kDa protein, chaperonin 10 [116] having immunosuppressive and growth factor properties [117], was identified. Chaperonin 10 belongs to the family of heat shock proteins but, unlike other members of this family, EPF is detected extracellularly [117]. Further experimental work is needed for the development of an accurate on-farm diagnostic test.

### 2.5. Current Developments in Early Pregnancy Diagnosis

During elongation of the blastocyst, trophectoderm cells secret interferon-tau (IFNT) into the uterine cavity. With its very low levels in extrauterine tissues and in the peripheral circulation, IFNT is regarded as a signal for the maternal recognition of bovine pregnancy. INFT contributes to the maintenance of the CL by blocking prostaglandin F_2α_ secretion of the endometrium. Currently, there is no accurate assay for diagnosing early pregnancy based on measuring IFNT concentrations. At the same time, the determination of interferon-tau stimulated genes (ISG) has been recently suggested as an alternative method for the indirect detection of the conceptus itself. According to a recent review [14], the relative abundance of ISG in total leukocytes, peripheral blood mononuclear cells and polymorphonuclear cells in pregnant cows from Days 18 to 20 after AI may be significantly higher than in non-pregnant cows.

Another biomarker for diagnosing early pregnancy is the measurement of circulating microRNAs; however, their use is currently limited to research investigations because standardized laboratory techniques are needed to isolate and measure them [14].

Proteomics analysis of the milk identified three possible biomarkers (lactoferrin, lactotransferrin, and alpha1G) for diagnosing early pregnancy [118], while proteomics analysis of the blood identified another three genes (Myxovirus resistance: MX1 and MX2 and oligoadenylate synthetase-1: OAS1), which can be used for early pregnancy diagnosis after validation on a large number of dairy cows [119]. Glycans may also play some critical roles in both the normal function of cells and in disease; therefore, bovine pregnancy can be predicted from a glycan biomarker present in a cow’s milk some 2–4 weeks earlier than by the standard method of pregnancy detection using ultrasonography [120]. Circulating nucleic acids (CNAs) or preimplantation factor (PIF) can be another biomarker for diagnosing early pregnancy in dairy cows [121,122], while Barbato et al. [123] suggested measuring PAG-2 mRNA in maternal blood cells, which can be detected earlier than the PAG-1 placental proteins in water buffalo and in other ruminants, as well.

These new branches of diagnostic sciences may contribute to finding molecules that may be exclusively related to maternal metabolic alterations during early embryonic development and to signaling for maternal recognition of pregnancy and continued survival [124], and may contribute to the development of an accurate early pregnancy diagnostic test in dairy cows.

A new technology (in-line milk analysis system, Herd Navigator) has already made possible the automatic collection of milk samples at milking robots or in the milking parlor to analyze progesterone, lactate dehydrogenase, and beta-hydroxybutyrate to detect estrus, tissue damage, and metabolic disorders, respectively [125]. According to Bruinjé and Ambrose [126], by using this new technology for early pregnancy diagnosis, a high sensitivity (>95%) could be reached from Day 27 after AI, while the specificity was somewhat lower before Day 40 after AI. After finding an accurate biomarker for early pregnancy diagnosis, its continuous measurements during milking will make it possible to diagnose pregnancy loss much earlier, meaning we can greatly contribute to increasing reproductive efficiency in our dairy herds. The importance of this technology would also be emphasized by its ability to identify pregnant and non-pregnant animals in a timely manner with no animal handling, because even a simple transrectal examination of dairy cows can lead to increased plasma and salivary cortisol concentrations and changes in heart rate, heart rate variability, and behavior that are indicative of pain [127].

## 3. Diagnosis of Pregnancy Losses (PL)

### 3.1. Ultrasonography

One of the advantages of UEs is that PL can be recognized by the absence of a heartbeat, the detachment of the fetal membranes, the appearance of particles in the fetal fluids, or the lack of the embryo proper [25,128]. UEs have revealed that LEM may occur in up to 23% of pregnancies [28,129]. PL (8%) diagnosed by ultrasonography in cows between Days 26 and 58 post-AI occurred at approximately Day 29 (n = 1), Day 33 (n = 3), Day 37 (n = 3), Day 40 (n = 2), Day 44 (n = 1), and Day 56 (n = 1) after AI. The exact day of occurrence of LEM/EFM could not be determined because UE was performed at intervals of 3–4 days [71].

According to Kelly et al. [130], decreased crown-rump length and progesterone concentration measured on Day 34 of gestation tended to be associated with an increased odds ratio for pregnancy loss, whereas CL perfusion and reduced blood flow of the uterine arteries evaluated by Doppler ultrasonography could not be used for predicting pregnancy loss in lactating dairy cattle.

### 3.2. Pregnancy Proteins

After diagnosing spontaneous cases of LEM by ultrasonography, both plasma bPSPB and bPAG-1 levels began to decline in most cases, while the CL continued to produce progesterone [63,71,72]. This confirms the previous observations [70,131], and demonstrates that lower progesterone concentrations are not the cause of conceptus death.

Although the concentrations of both proteins decrease steadily [70,71] after spontaneous or induced LEM/EFM, they reach the threshold level only after a relatively long half-life, namely, about 7 to 8 days for the bPSPB RIA test [70], and 3 to 4 days for the bPAG-1 RIA test [72]. Thus, they can contribute to the elevation of false positive pregnancy diagnoses on the farm. Similar results were reported by Giordano et al. [63] when inducing LEM on Day 39 of gestation and using the PSPB commercial and the PAG in-house ELISA tests. Although the threshold levels for these tests were not determined, they reached the basal levels in both tests approximately 5 to 7 days after inducing LEM. Based on commercial blood and milk ELISA tests, the threshold levels are reached approximately 7 to 14 days after pregnancy loss [37].

According to Mercadante et al. [81], reduced PAG concentrations (in-house ELISA) at Day 32 after AI may predict pregnancy loss between Days 46 and 74 of gestation. Based on positive and negative predictive value analysis, a circulating concentration of PAG (in-house ELISA) below 1.4 ng/mL in lactating dairy cattle following timed AI and 1.85 ng/mL following timed embryo transfer on Day 7 was 95% accurate in predicting LEM/EFM (between Days 31 and 59) at Day 31 of gestation [13]. It has been recently reported that cows being pregnant at Day 31 of gestation and maintaining the pregnancy until Day 59 had significantly higher circulating concentrations of PAG (commercial ELISA test) at Day 31 of gestation compared with cows that experienced LEM/EFM between Days 31 and 59 of gestation [37]. In contrast, although there was a significant difference in the PAG concentrations measured on Day 24 after AI between pregnant and non-pregnant multiparous cows while in heifers only a tendency was detected at Day 31 after AI, the circulating concentrations of PAG at Day 24 of gestation in animals that maintained pregnancy until Day 60 compared to animals that lost pregnancy between Days 31 and 60 of gestation, were not significantly different [132].

López-Gatius et al. [133] reported that low or very high plasma pregnancy-associated glycoprotein-1 (PAG-RIA) levels on Day 35 of gestation in cows were related to a subsequent pregnancy loss. Similarly, Gábor et al. [134] also found pregnancy losses in cows with high PSPB (commercial ELISA) concentrations (>1.1ng/mL) and in cows with low concentrations of both PSPB and progesterone (0.6–1.1 and <2ng/mL, respectively), however the prevalence of pregnancy loss was significantly lower in cows with high PSPB concentrations (15%) between Days 29 and 35 of gestation than in those with low concentrations (76.3%).

It has been recently reported that different PAG ELISA assays may accurately detect pregnancy; however, the ability to predict embryo survival vs. mortality during early gestation appears to be antibody-/assay-dependent [135].

In order to be able to decrease the effect of false positive diagnoses, due to pregnancy loss on the farm, it is necessary to repeat the pregnancy tests [136]. By using commercial plasma or milk ELISA tests, the optimal time for the first pregnancy diagnosis is around Day 32 after AI, when plasma and milk PAG levels are at an early peak. After this period, all pregnant cows should be retested on Day 74 after AI or later, when plasma and milk PAG levels rebound from their nadirs [37]. In contrast to the gradual increase in PAG-1 concentration (RIA) throughout gestation [61], plasma and milk PAG levels (commercial ELISA) reached a peak at Day 32 of gestation and then started to decrease to a nadir from Days 53 to 60 for the plasma PAG level and from Days 46 to 67 for the milk PAG level, followed by a gradual increase in PAG levels from Days 74 to 102 after AI [37].

The potential clinical significance of diagnosing pregnancy loss using ultrasonography or pregnancy protein tests and treating the cows with prostaglandin as soon as possible is that these measures may reduce the number of days before re-insemination [137].

## 4. Future Perspectives

Pregnancy diagnosis plays an essential role in decreasing days open in dairy farms; therefore, it is very important to select an accurate method for diagnosing early pregnancy, because the cost of each day open past 100 DIM may reach $4.00 [138] or €2.5 to 6.5 [139], respectively. Besides traditional pregnancy diagnoses made by rectal palpation of the uterus from 40 to 60 days after AI and measuring the serum or milk progesterone concentration between 18 to 24 days after AI [14,140,141], there are several new possibilities to diagnose early pregnancy in dairy farms; however, it is very important to emphasize that before introducing any new diagnostic test we need to make sure the accuracy of that particular test to be able to decrease the rate of iatrogenic pregnancy losses caused by prostaglandin treatment to reduce the interval to the next AI service [140] or resynchronization of the cows [142,143]. Furthermore, the new pregnancy diagnostic results must be confirmed by the old diagnostic method to decrease the negative effects of false negative diagnoses [144]. Linear-array/sector B-mode [145] and Doppler ultrasonography [14] may exceed the other diagnostic methods in the amount of information collecting from each animal during scanning, however, their uses greatly depend on the operator proficiency and availability [145].

## 5. Conclusions

The successful genetic selection for higher milk production caused a dramatic decline in the reproductive performance of dairy cows all over the world during the last decades. Achievement of optimum herd reproductive performance (calving interval of 12 or 13 months with the first calf born at 24 months of age) requires concentrated management activities especially during calving and during the first 100 days following calving. One of the most important management activities needed to pursue during the early postpartum period to reach or approach the optimal reproductive performance is diagnosis of early pregnancy diagnosis and embryonic mortality in dairy cows. There are several diagnostic methods available for the dairy farms such as rectal palpation, transrectal ultrasonography, chemical and hormone assays, however, transrectal B-mode and Doppler ultrasonography may exceed the other diagnostic methods in the amount of information collecting from each animal during scanning in the farm. The advantages and disadvantages of the different diagnostic methods were discussed in order to be able to select the most accurate method for the dairy.

## Figures and Tables

**Table 1 animals-11-01666-t001:** Identifiable characteristics of the bovine conceptus during ultrasonographic examinations for pregnancy diagnosis, indicating the first day of detection.

Characteristics	First day of Detection in Heifers Curran et al. [21] Range	First Day of Detection in Heifers Curran et al. [21] Mean ± SEM	First Day of Detection after ET Totey et al. [22] Mean ± SD
Embryo proper	19 to 24	20.3 ± 0.3	19.5 ± 0.7
Heartbeat	19 to 24	20.9 ± 0.3	22.6 ± 0.9
Allantois	22 to 25	23.2 ± 0.3	23.1 ± 0.8
Spinal cord	26 to 33	29.1 ± 0.5	33.0 ± 1.5
Forelimb buds	28 to 31	29.1 ± 0.3	32.7 ± 1.3 *
Amnion	28 to 33	29.5 ± 0.5	25.1 ± 1.4
Hindlimb buds	30 to 33	31.2 ±0.3	32.9 ± 1.3 **
Placentomes	33 to 38	35.2 ± 1.0	-
Split hooves	42 to 49	44.6 ± 0.7	-
Fetal movement	42 to 50	44.8 ± 0.8	50.7 ± 1.0
Ribs	51 to 55	52.3 ± 0.5	60.9 ± 1.7

ET: embryo transfer, * forelimb, ** limb buds.

**Table 2 animals-11-01666-t002:** Accuracy of ultrasonographic examinations for diagnosing early pregnancy in dairy cattle by using a 5.0 MHz linear-array or sector transducer.

Days after AI	N	Sensitivity (%)	Specificity (%)	Positive Predictive Value (%)	Negative Predictive value (%)	References
22–40	435	96.2	71.1	89.6	87.8	Filteau and DesCoteaux, [26]
25–29	101	98	91.3	93	97.6	Szenci et al. [27] ^a^
30–39	143	100	100	100	100	
26 ^b^	48	100	96.7	94.4	100	Romano et al. [17]
27 ^b^	53	100	92.8	92.6	100	
26–33	85	97.7	87.8	89.6	97.2	Pieterse et al. [15]
29 ^c^	151	100	96.3	91.3	100	Romano et al. [17]
30 ^c^	151	100	97.4	91.9	100	
30–39	722	95	67	89	84	Hanzen and Laurent [28]
40–49	620	98	77	92	94	
31–35	323	98	80	85.7	97	Munoz del Real et al. [29]
36–40		97	93	93.6	97	

^a^ Sector transducer, ^b^ Heifers, ^c^ Cows, NG: not given. Sensitivity = correct positive/correct positive + false negative × 100. Specificity = correct negative/correct negative + false positive × 100. Positive predictive value = correct positive/correct positive + false positive × 100. Negative predictive value = correct negative/correct negative + false negative × 100.

**Table 3 animals-11-01666-t003:** Accuracy of ultrasonographic examinations for diagnosing early pregnancy in dairy cattle by using a high frequency linear-array transducer.

Days after AI	N	Sensitivity (%)	Specificity (%)	Positive Predictive Value (%)	Negative Predictive Value (%)	References
27	1673	96.5	93.4	89.7	97.8	Silva et al. [18] (5–10 MHz transducer)
28	100	92.7	91.5	88.4	94.7	Karen et al. [32] (6–10 MHz transducer)
28–35	497	96 ^a^	83	91	92	Nation et al. [16] (7.5 MHz transducer)
		97 ^b^	86	92	93	
29–30	138	90.4 ^c^	96.0	95.0	92.3	Szenci et al. [31] (7.5 MHz transducer)
33–34	135	96.6 ^c^	98.6	98.3	97.3	
33–34	135	90.0 ^d^	100	100	92.5	
30	47	92.3	97.1	92.9	97.1	Abdullach et al. [33] (6.5 MHz transducer)

^a^ Observation of ≥15 mm fluid in the uterine lumen and embryonic membranes; ^b^ Observation of an embryo with beating heart; ^c^ Recognition of allantoic fluid was used as the criterion for a positive pregnancy diagnosis; ^d^ Recognition of an embryo proper with a beating heart was used as the criterion for a positive pregnancy diagnosis.

**Table 4 animals-11-01666-t004:** Accuracy of in-house early-pregnancy associated glycoproteins (PAGs) ELISA tests for diagnosing early pregnancy in dairy cattle.

Days after AI	N	Type of Sample	Sensitivity (%)	Specificity (%)	Positive Predictive Value (%)	Negative Predictive Value (%)	References
26–58	169	Serum	97.8	91.2	97.8	91.2	Piechotta et al. [78] ^a^
27–35	2129	Plasma	95.1	89.0	90.1	94.5	Sinedino et al. [79] ^b^
31–35	209		98.7	88.1	83.7	99.1	
26–30	106	Serum *	94	77	NG	NG	Friedrich and Holtz [77]
31–35	88		95	93	NG	NG	
36–40	57		95	93	NG	NG	
>40	128		100	92	NG	NG	
27	1673	Plasma	95.4	94.2	90.7	97.1	Silva et al. [18] ^b^
28	100	Plasma	95.3	88.3	NG	NG	Thompson et al. [80] ^b^
30	100	Plasma	100	90.9	NG	NG	

* using a threshold of 2 ng/mL; ^a^ Used an in-house ELISA assay described by Friedrich and Holtz [77]; ^b^ Used an in-house ELISA assay described by Green et al. [76] (2005); NG: not given.

**Table 5 animals-11-01666-t005:** Accuracy of commercially available ByoPRYN pregnancy ELISA tests for diagnosing early pregnancy in dairy cattle.

Days after AI	N	Type of Sample	Sensitivity (%)	Specificity (%)	Positive Predictive Value (%)	Negative Predictive Value (%)	References
23	544	Serum	92.8	97.0	NG	NG	Martins et al. [83]
23 ^a^			98.0	97.0			
28			100	91.6			
24	206	Serum	100 ^b^	93.6 ^b^	93.3 ^b^	100 ^b^	Middleton and Pursley [84]
26–58	185	Serum	98.0	97.1	99.3	91.7	Piechotta et al. [78]
28	246	Plasma	93.9	95.5	94.7	94.7	Romano and Larson [85]
30	229		96.0	93.9	92.2	96.8	
35	246		97.2	93.6	92.0	97.8	
28–41	507	Serum	97.0	95.9	96.2	96.7	Breed et al. [86]
35–60	976		96.6	92.2	95.0	94.7	
30–36	1742	Serum	95.1	68.6	NG	NG	Gábor et al. [87]
30–36	1336	Serum	100	87.8	NG	NG	Howard et al. [88]

^a^ A cow was considered pregnant when there was an increase in serum PSPB from basal (Day 17) to Day 23 after AI of more than 28%; ^b^ Only non-pregnant animals were evaluated.

**Table 6 animals-11-01666-t006:** Accuracy of commercially available D29^®^ bovine blood pregnancy ELISA tests for diagnosing early pregnancy in dairy cattle.

Days after AI	N	Type of Sample	Sensitivity (%)	Specificity (%)	Positive Predictive Value (%)	Negative Predictive Value (%)	References
28	100	serum	90.2	98.3	97.4	93.7	Karen et al. [31]
29–36	202	blood	99.4	66.7 *	92.6	96.3	Paré et al. [89]
30–40	212	plasma	100	81.3	91	100	Moussafir et al. [90]

* 100% in non-inseminated cows.

**Table 7 animals-11-01666-t007:** Accuracy of commercially available IDEXX bovine pregnancy ELISA tests for diagnosing early pregnancy in dairy cattle.

Days after AI	N	Type of Sample	Sensitivity (%)	Specificity (%)	Positive Predictive Value (%)	Negative Predictive Value (%)	References
25	61	plasma	92.0	91.6	88.4	94.2	Kaya et al. [82]
28	84		97.1	100	100	98.0	
32	86		97.4	91.4	90.4	97.7	
25–32	231		95.9	94.7	93.1	96.9	
28	210	serum	100	94.9	94.4	100	Akkose et al. [91]
28	320	plasma	97 *	94 **	95 **	97 *	Mayo et al. [92]
30 ± 1	116	plasma	100	88.9	93.8	100	Commun et al. [93]
30 ± 1	116	serum	100	88.6	93.8	100	
32	141	plasma	100	87	84	100	Ricci et al. [37]
41 ± 2	116	plasma	100	100	100	100	Commun et al. [93]
41 ± 2	116	serum	98.4	96.8	98.4	96.8	

* ±1, ** ±2.

**Table 8 animals-11-01666-t008:** Accuracy of commercially available milk pregnancy ELISA tests for diagnosing early pregnancy in dairy cattle.

Days after AI	N	Sensitivity (%)	Specificity (%)	Positive Predictive Value (%)	Negative Predictive Value (%)	References
28	320	96 *	94 *	94 *	96 *	Mayo et al. [92]
28–45	497	99	95	NG	NG	Dufour et al. [38]
28–35	1006	99.2	93.4	NG	NG	Fosgate et al. [34]
30 ± 1	116	98.1	90.3	94.5	96.6	Commun et al. [92]
32	135	98	83	79	99	Ricci et al. [37]
33–52	119	100	97.9	98.5	100	Lawson et al. [94]
41 ± 2	116	100	92.3	96.4	100	Commun et al. [93]
53 ± 2	116	98.1	96.2	98.1	96.2	
≥60	683	99.2	95.5	99.8	80.8	LeBlanc [95]
≥60	602	98.5	96.7	NG	NG	Byrem et al. [96]

NG: not given, * ±2.

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
