# Peer review of "Recent Possibilities for the Diagnosis of Early Pregnancy and Embryonic Mortality in Dairy Cows"

_animals, 2021, doi:10.3390/ani11061666_

Round 1

Reviewer 1 Report

Revision of the manuscript entitled ‘ Recent possibilities for the diagnosis of early pregnancy and  embryonic mortality in dairy cows’ by Szenci

The paper is an interesting revision of the current diagnostic methods of pregnancy in cattle. I have some suggestions before publishing in ANIMALS

In the introduction section, the objective of the review is needed.

In my opinion, the PD by ultrasound section is quite confusing. We need to differentiate heifers from mature cows, and discuss something about twins. In each case, the days and precision of the ultrasound are different. Please, try to arrange this section.

Can you please add a new section in Doppler ultrasonography discussing about uterine blood flow in pregnant animals?

The conceptus proteins section is quite confusing. Can you please add more subsections and clarify it?

The PL section is quite confusing again. In my opinion, you should talk about: US, PAG or PSPB and Progesterone. Because now is all mixed.

Some conclusions are not conclusions at all. Maybe you can add future perspectives or something similar

Author Response

Responses to Reviewer Comments:

The author is grateful for the efforts of Reviewer I in the evaluation of his manuscript. I appreciate your time spent with the review. Before answering the most important concerns, let me thank you for your valuable comments on the paper. I feel that Reviewer I’s comments and recommendations were reasonable, and I tried to take them into account as far as possible while improving the manuscript. In my opinion, the activities of the reviewers have contributed significantly to the improvement of the quality of my paper. As you will see, I have made all the corrections required.

Reviewer I Comments

Revision of the manuscript entitled ‘Recent possibilities for the diagnosis of early pregnancy and  embryonic mortality in dairy cows’ by Szenci

The paper is an interesting revision of the current diagnostic methods of pregnancy in cattle. I have some suggestions before publishing in ANIMALS

Rew#1: In the introduction section, the objective of the review is needed.

AU: Yes, it was corrected: The PL represents a considerable biological and economic waste for the farmer therefore the objective of the present review is to discuss recent methods suitable for diagnosing early pregnancy and pregnancy losses in dairy cattle.

Rew#1: In my opinion, the PD by ultrasound section is quite confusing. We need to differentiate heifers from mature cows, and discuss something about twins. In each case, the days and precision of the ultrasound are different. Please, try to arrange this section.

AU: Please note that according to my best knowledge there was only one publication that evaluated the accuracy of PD for heifers and cows separately and this is mentioned in the manuscript. Regarding the diagnosis of twin pregnancy, it has just been published by us in Animals therefore it would be not good to repeat it again.

Rew#1: Can you please add a new section in Doppler ultrasonography discussing about uterine blood flow in pregnant animals?

AU: Due to the fact that it would be very complicated to use Doppler ultrasonography for PD in the field therefore presently I cannot suggest this method to use.  It is also important to mention that according to Kelly et al. (2017) CL perfusion and uterine blood flow were even not associated with an increased odds ratio of pregnancy loss at Day 34 of gestation therefore it cannot be suggested to predict PL. This reference was added to the manuscript.

Rew#1: The conceptus proteins section is quite confusing. Can you please add more subsections and clarify it?

AU: According to your recommendation I have tried to clarify this section by giving subsections.

Rew#1: The PL section is quite confusing again. In my opinion, you should talk about: US, PAG or PSPB and Progesterone. Because now is all mixed.

AU: According to your recommendation I have tried to clarify this section and the following was added:

According to Kelly et al. [130], decreased crown-rump length and progesterone concentration measured on Day 34 of gestation tended to be associated with an increased odds ratio for pregnancy loss, whereas CL perfusion and reduced blood flow of the uterine arteries evaluated by Doppler ultrasonography could not be used for predicting pregnancy loss in lactating dairy cattle.

Rew#1: Some conclusions are not conclusions at all. Maybe you can add future perspectives or something similar.

AU: According to your recommendation future perspectives were used.

Reviewer 2 Report

Dear author,

This submitted manuscript is well written and provides an useful, extensive but also comprehensive overview as clearly is covered by the Title of this manuscript. Indeed, recent possibilities have been added to the list of "old" possibilities to correctly diagnose early pregnancy and hence embryonic mortality in dairy cows. In this respect I completely agree with the 'warning' as is stated in the section 4-Conclusions as cited: "the new pregnancy diagnostic results must be confirmed by the old diagnostic method to decrease the negative effects of false negative diagnoses" (lines 430-431).

Therefore, I do not have any added comments to this manuscript and qualify this version as "accepted without further review". 

Author Response

Responses to Reviewer Comments:

The author is grateful for the efforts of Reviewer II in the evaluation of his manuscript. I appreciate your time spent with the review. Before answering the most important concerns, let me thank you for your valuable comments on the paper. I feel that Reviewer II’s comments and recommendations were reasonable, and I tried to take them into account as far as possible while improving the manuscript. In my opinion, the activities of the reviewers have contributed significantly to the improvement of the quality of my paper. As you will see, I have made all the corrections required.

Reviewer II Comments

Dear author,

This submitted manuscript is well written and provides an useful, extensive but also comprehensive overview as clearly is covered by the Title of this manuscript. Indeed, recent possibilities have been added to the list of "old" possibilities to correctly diagnose early pregnancy and hence embryonic mortality in dairy cows. In this respect I completely agree with the 'warning' as is stated in the section 4-Conclusions as cited: "the new pregnancy diagnostic results must be confirmed by the old diagnostic method to decrease the negative effects of false negative diagnoses" (lines 430-431).

Therefore, I do not have any added comments to this manuscript and qualify this version as "accepted without further review". 

AU: The author would like to thank Reviewer II for finding merit in the manuscript.

Reviewer 3 Report

This was a very well-detailed and needed review of the science of early pregnancy diagnoses in dairy cattle. The manuscript was very well written. The author was meticulous in describing a very in-depth examination of the literature as it relates to early pregnancy diagnoses.  This review will have a significant impact in the literature. With that being said, there two aspects of this manuscript that must be revised in order for this review to be acceptable.  1) Remove all mentions of using P4 or color Doppler blood flow near the time of return to estrus (18 to 24 d post AI) as an accurate way to diagnose pregnancy. It clearly is not an accurate determination of pregnancy. The author should either review this literature and report the work that clearly shows the inaccuracy of pregnancy diagnosis using P4 or color Doppler, or simply remove all statements in this regard. 2) The author was redundant in reporting outcomes that were in Tables and in the text. One example: “Between Days 27 to 34 after AI, the sensitivity, specificity, positive predictive value 99 and negative predictive value of ultrasonographic diagnoses made by the use of a 5- to 100 10-MHz transducer were 90% to 96.6%, 91.5% to 100%, 88.4% to 100% and 92.3% to 97.8%, 101 respectively (Table 3).” Please be more general in describing tables in the text.

Author Response

Responses to Reviewer Comments:

The author is grateful for the efforts of Reviewer III in the evaluation of his manuscript. I appreciate your time spent with the review. Before answering the most important concerns, let me thank you for your valuable comments on the paper. I feel that Reviewer III’s comments and recommendations were reasonable, and I tried to take them into account as far as possible while improving the manuscript. In my opinion, the activities of the reviewers have contributed significantly to the improvement of the quality of my paper. As you will see, I have made all the corrections required.

Reviewer III Notes

This was a very well-detailed and needed review of the science of early pregnancy diagnoses in dairy cattle. The manuscript was very well written. The author was meticulous in describing a very in-depth examination of the literature as it relates to early pregnancy diagnoses.  This review will have a significant impact in the literature. With that being said, there two aspects of this manuscript that must be revised in order for this review to be acceptable. 

Rew#III: 1) Remove all mentions of using P4 or color Doppler blood flow near the time of return to estrus (18 to 24 d post AI) as an accurate way to diagnose pregnancy. It clearly is not an accurate determination of pregnancy. The author should either review this literature and report the work that clearly shows the inaccuracy of pregnancy diagnosis using P4 or color Doppler, or simply remove all statements in this regard.

AU: It has been changed as it was requested.

Rew#III: 2) The author was redundant in reporting outcomes that were in Tables and in the text. One example: “Between Days 27 to 34 after AI, the sensitivity, specificity, positive predictive value 99 and negative predictive value of ultrasonographic diagnoses made by the use of a 5- to 100 10-MHz transducer were 90% to 96.6%, 91.5% to 100%, 88.4% to 100% and 92.3% to 97.8%, 101 respectively (Table 3).” Please be more general in describing tables in the text.

AU: Yes, it is. However, it would be very important to give the range of the different values. Therefore I have changed the sentences in each case in the text: ….varied between 90% and 96.6%, 91.5% and 100%, 88.4% and 100%, and 92.3% and 97.8%, respectively (Table 3).

Round 2

Reviewer 1 Report

For me, the article can be published in the present form

Reviewer 3 Report

Any reference to using progesterone concentrations for pregnancy diagnosis is absolutely false and ridiculous.